# Simulated Fermentation of Strong-Flavor Baijiu through Functional Microbial Combination to Realize the Stable Synthesis of Important Flavor Chemicals

**DOI:** 10.3390/foods12030644

**Published:** 2023-02-02

**Authors:** Youqiang Xu, Mengqin Wu, Dong Zhao, Jia Zheng, Mengqi Dai, Xiuting Li, Weiwei Li, Chengnan Zhang, Baoguo Sun

**Affiliations:** 1School of Food and Health, Beijing Technology and Business University (BTBU), Beijing 100048, China; 2Wuliangye Yibin Co., Ltd., Yibin 644000, China; 3Key Laboratory of Brewing Microbiome and Enzymatic Molecular Engineering, China General Chamber of Commerce, Beijing 102401, China; 4Key Laboratory of Brewing Molecular Engineering of China Light Industry, Beijing Technology and Business University (BTBU), Beijing 100048, China

**Keywords:** Strong-flavor Baijiu, flavor esters, microbial composition, correlation analysis, microbial interaction

## Abstract

The solid-state fermentation of Baijiu is complicated by the co-fermentation of many microorganisms. The instability of the composition and abundance of the microorganisms in the fermentation process leads to fluctuations of product quality, which is one of the bottleneck problems faced by the Strong-flavor Baijiu industry. In this study, we established a combination of functional microorganisms for the stable fermentation of the main flavor compounds of Baijiu, including medium and long-chain fatty acid ethyl esters such as hexanoic acid, ethyl ester; butanoic acid, ethyl ester; octanoic acid, ethyl ester; acetic acid, ethyl ester; 9,12-octadecadienoic acid, ethyl ester; and decanoic acid, ethyl ester in the fermented grains. Our study investigated the effects of microbial combinations on the fermentation from three aspects: microbial composition, microbial interactions, and microbial association with flavor compounds. The results showed that the added functional microorganisms (*Lactobacillus*, *Clostridium*, *Caproiciproducens*, *Saccharomyces*, and *Aspergillus*) became the dominant species in the fermentation system and formed positive interactions with other microorganisms, while the negative interactions between microorganisms were significantly reduced in the fermentation systems that contained both Daqu and functional microorganisms. The redundancy analysis showed that the functional microorganisms (*Lactobacillus*, *Saccharomyces*, *Clostridium*, *Cloacibacterium*, *Chaenothecopsis*, *Anaerosporobacter*, and *Sporolactobacillus*) showed strong positive correlations with the main flavor compounds (hexanoic acid, ethyl ester; lactic acid, ethyl ester; butanoic acid, ethyl ester; acetic acid, ethyl ester; and octanoic acid, ethyl ester). These results indicated that it was feasible to produce Baijiu with a functional microbial combination, and that this could promote stable Baijiu production.

## 1. Introduction

Baijiu is one of the national beverages in China with a long history of thousands of years and is renowned overseas [1,2]. Among Baijiu, Strong-flavor Baijiu is the dominant product of the Baijiu industry, with sales accounting for about 70% of the market share. The raw materials for manufacturing Strong-flavor Baijiu mainly include sorghum [1,2]. The raw materials are moistened with water, and mixed with rice husks and steamed thereafter (Figure 1). Then the fermented grains are cooled to room temperature, and a certain amount of Daqu powder is added and mixed evenly, and the mixtures are transferred to the container (usually called “Jiaochi” in Chinese) for solid-state fermentation with a time period of about 30–60 days [3]. Finally, the Baijiu products are obtained by distillation (Figure 1).

Studies indicate that the metabolism of microorganisms in the solid-state fermentation process plays an important role in the transformation of raw materials [2]. Meanwhile, microbial metabolism can not only synthesize many kinds of flavor chemicals, but also degrade some potentially harmful substances in the raw materials, and so greatly affect the quality of the products [4,5,6,7]. Daqu, the fermentation starter of Baijiu, is rich in microorganisms, and has a significant impact on the Baijiu fermentation [8]. However, due to the traditional, undeveloped manufacturing process of Daqu, the composition and abundance of microorganisms in different batches of Daqu are not completely consistent, leading to the quality divergence of Daqu [9,10,11], which has a great impact on the microbial fermentation of Baijiu. The production of high-quality Baijiu products shows a dependence on high-quality Daqu [12]. In addition, air, water, tools, operators, and microorganisms in pit mud will participate in the fermentation process [2]. Thus the microbial composition in the fermentation process of Strong-flavor Baijiu is relatively complex and unstable, and the instability of the microbial composition generates the instability in the quality of fermentation products among batches, which is one of the key problems to be solved in the Strong-flavor Baijiu industry [2,13]. Therefore, whether we could control the complex microbial system, and realize a relatively stable fermentation, is the problem addressed in this work.

The quality of Baijiu products is closely related to the composition of flavor chemicals [14]. According to the national standard (GB/T10781.1-2021), Strong-flavor Baijiu is one kind of product with esters as the main flavor chemicals, and among which fatty acid esters show relatively high concentrations and low odor threshold values. Previous studies have also investigated the microbial metabolism of relevant flavor compounds. There are also studies about ester synthesis, mainly including two metabolic pathways, one is the transesterification reaction by yeast species, and the other is the esterification of acids and alcohols (mainly ethanol, produced by functional microorganisms such as *Saccharomyces cerevisiae*) [2,4]. The metabolic synthesis of fatty acids is closely related to bacteria, such as *Clostridium* spp. (e.g., *C. butyricum* and *C. kluyveri*) identified from the pit mud, which are closely related to the production of butyric acid and caproic acid, respectively [15,16]. Early studies found that fungi such as *Monascus purpureus* and *Aspergillus niger* isolated from Baijiu can synthesize esters through esterification with acids and ethanol as substrates [4,6,17]. According to the national standard of Strong-flavor Baijiu, the relevant functional microorganisms could be recognized as the important core microorganisms in the complex microbial fermentation system.

Recent studies about complex microbial co-fermentation systems show that the core microbial composition can drive the whole fermentation process, and plays an important role in maintaining the stability of the fermentation system, as well as affecting the flavor and quality of the products [13,18,19]. Driving the whole fermentation process with the core microbial composition is recognized as helpful for maintaining the stability of microbial co-fermentation, and is also applicable in the Baijiu industry [13,20]. Our previous work also proposed four criteria for identifying the core microorganisms from Baijiu, including (1) microorganisms hydrolyzing raw materials such as starch and protein, (2) microorganisms producing flavor chemicals, (3) microorganisms promoting the synthesis of flavor chemicals by other microorganisms, and (4) microorganisms coordinating the coexistence of other microorganisms [2]. Among these microorganisms, the strains hydrolyzing raw materials and producing flavor chemicals are particularly important. As studies about microorganisms of Baijiu are still in progress, many functional microorganisms are just being discovered or remain unidentified [21,22,23]. Therefore, based on previous work, and criteria for functional microorganisms of Baijiu, a microbial combination was proposed, which included functional fungal strains for raw material conversion and flavor ester synthesis, as well as ethanol producing yeast strains and acid producing bacterial strains. Their performance in Baijiu fermentation was investigated in this work. Results showed that the stable production of Baijiu could be realized by using this combination of functional microorganisms (without adding Daqu). In addition, the combination of functional microorganisms and Daqu, for simulated solid-state fermentation, could achieve the stable synthesis of important flavor esters in Baijiu by comparing the fermentation systems of the three experimental groups. This confirmed that the use of a functional microbial combination could reduce the dependence on Daqu and improve the stability of production, which provided a new direction for reducing the cost and improving the quality of Baijiu.

## 2. Materials and Methods

### 2.1. Sample Preparation and Collection

The simulated fermentation systems contained different fermentation starters, including the control sample (without the addition of Daqu or microorganisms), sample 2 (with added Daqu, starter 2 in Figure 2), sample 3 (with added microbial combination, starter 3 in Figure 2), and sample 4 (with added Daqu and microbial combination, starter 4 in Figure 2). The Daqu was obtained from a company in Sichuan province, China. All the functional microbes were obtained from the China General Microbiological Culture Collection Center (CGMCC) or laboratory collection (Table 1). The strains were cultivated and collected by centrifuging at 6000× *g* and 4 °C for 10 min, washed twice by sterile water, and the cell density of the bacterial suspension of all strains was adjusted to 1.0 at OD_600nm_.

#### 2.1.1. Sample Preparation

Raw materials were mixed according to the ratio of crushed grain: whole grain of 1:9, washed twice, and then the sorghum grains were steamed at 115 °C for 40 min. Water was added to the steamed grain to adjust the moisture to between 50 and 53%. After this the temperature was reduced to 20 °C, and 20% rice husk was added to the grains and mixed evenly. As shown in Figure 2, sample 1 was designed as the negative control, without the addition of Daqu powder or microorganisms; sample 2, with 20% Daqu powder added to the raw materials; sample 3 with 20% functional microorganism combination added to the raw materials; and sample 4 with 10% Daqu powder and 10% microorganism combination added.

#### 2.1.2. Sample Collection

The samples were collected at 7-day intervals in duplicate, one for the flavor component analysis and physicochemical factors analysis, and the other for the determination of the microbial composition, as shown in Table 2. All samples were kept at –80 °C.

### 2.2. Determination of Physicochemical Factors

Five physicochemical factors were detected during fermentation, including moisture, acidity, pH value, starch content, and reducing sugars content. The moisture of the samples was determined using the gravimetric method, by drying the samples at 105 °C for at least 3 h. The acidity of the samples was analyzed by acid-base titration with NaOH (0.1 M) and using phenolphthalein as an indicator (endpoint of pH 8.2) [24]. The pH value of the samples was investigated using a pH meter [24]. The Fehling reagent method was used to determine the content of starch and reducing sugars, using methylene blue as an indicator [25]. Operations were carried out according to the relevant standard, T/CBJ 004-2018.

### 2.3. Analysis of Volatile Flavor Compounds in Fermented Grains

Saturated sodium chloride solution (5 mL) was added to the fermented grains (2 g) in a 20 mL bottle. The internal standard 4-octanol was added, with a concentration of 4 mg/L, followed by ultrasonicating for 10 min. The volatile compounds in the fermented grains were assayed by headspace solid-phase microextraction coupled with gas chromatography-mass spectrometry (HS-SPME-GC-MS) (TSQ 8000 Evo, Trace MS/GC, Thermo Fisher Scientific, Waltham, MA, USA) equipped with a flame ionization detector. The volatiles of the fermented grains were extracted with a 50/30 μm DVB/CAR/PDMS fiber (Supelco, Bellefonte, PA, USA), and after equilibrating at 60 °C for 20 min, a fiber was used to extract the volatiles for 30 min, followed by desorbing at the inlet for 5 min, following our previous studies [26,27]. The initial column temperature was kept at 50 °C for 2 min, followed by increasing it to 85 °C at a rate of 2 °C/min, and then kept at 85 °C for 5 min, and then increased to 150 °C at a rate of 5 °C/min and maintained there for 10 min, and finally increased to 250 °C at a rate of 5 °C/min and kept there for 20 min. The flow rate of the helium carrier gas was 1 mL/min [26]. The column was a TG-5MS column (30 m × 0.25 mm × 0.25 μm, J&W Scientific, Folsom, CA, USA). Mass spectrometry (MS) was generated with an electron impact ionization energy of 70 eV and a full scan range from 30 to 400 amu [26]. The flavor chemicals were identified by matching the spectra to the NIST05 spectrum database.

### 2.4. Microbial Community Analysis

Metagenomic DNA from fermented grains was extracted using the E.Z.N.A.^®^ soil DNA kit (Omega Bio-tek, Norcross, GA, U.S.). The bacterial 16S rRNA gene was amplified with the primer pair 338F (5′-ACTCCTACGGGAGGCAGCAG-3′) and 806R (5′-GGACTACHV GGGTWTCTAAT-3′). The fungal 18S rRNA gene was amplified using primers ITS1 (5′-TCCGTAGGTGAACCTGCGG-3′) and ITS2 (5′-GCTGCGTTCTTCATCGATGC-3′). Amplicons were then sequenced using an Illumina MiSeq platform (Illumina, San Diego, CA, United States) at AuwiGene Technology Co., Ltd., Beijing, China.

### 2.5. Data Analysis

Heatmap analysis was performed using TBtools (version 1.1043). Spearman’s correlation coefficients for microbial interactions were calculated using R software (version 4.2.1), and correlations with |RHO| ≥ 0.6 and *p* < 0.05 were visualized by ChiPlot. The Spearman’s correlation coefficient was also used to determine the correlation analysis between dominant microorganisms and all flavor compounds in R software (version 4.2.1). Thereafter, the correlation heatmap was performed by the pheatmap package. The significant correlation was expressed by “*” (* *p* <0.05, ** *p* < 0.01, *** *p* < 0.001).

## 3. Results and Discussion

The purpose of this work was to construct the combination of core functional microorganisms for simulating fermentation to achieve the stable production of flavor compounds in Baijiu. Studies were performed by combining high-throughput sequencing technology, interactions between microorganisms, and the correlation between the microorganisms and the flavor compounds.

### 3.1. Analysis of Physicochemical Factors

Physicochemical factors can be used to judge the process of fermentation. The dynamic changes of physicochemical factors of all samples were detected during the 42 days of fermentation, including acidity, pH value, moisture, starch, and reducing sugars content. Compared with the control group, the moisture contents of the three experimental groups increased rapidly in the early and middle stages of fermentation, and gradually stabilised in the later stage, with an increase of about 11% (48.8% ± 0.2%–55.9% ± 1.3%; 47.0% ± 0.3%–54.3% ± 0.8%; and 48.3% ± 1.0%–59.7% ± 0.2%, in groups 2, 3, and 4, respectively) (Figure 3c), which indicated that microbial metabolism was carried out normally in each experimental group [28,29]. During fermentation, the acidity gradually increased, accompanied by a decrease in the pH value. According to Figure 3a, the acidity of the experimental groups increased rapidly during the period of fermentation from 14–28 days. In particular, the rising acidity in samples 2 and 4 was higher than that of sample 3, and this divergence was possibly caused by the differences of microbial community compositions (e.g., *Lactobacillus* could produce lactic acid, strongly associated with acidity, and its abundance in sample 3 was lower than those of samples 2 and 4,), and generated different metabolites. At the same time, the pH values of all samples decreased from 4.2 to about 3.7 during fermentation (Figure 3b). The synergism of multiple microorganisms hydrolyzed starch into reducing sugars, and then converted these into ethanol and various flavor compounds during Baijiu fermentation [28]. Therefore, the consumption of starch and reducing sugars, to a certain extent, reflected the process of fermentation [28]. As shown in Figure 3e, the starch content was highest at the initial stage of fermentation and gradually decreased through the process of fermentation, but the starch consumption of each sample was different at the end of the fermentation process. The starch consumption of sample 4 reached 13.8%, followed by sample 3 and sample 2, which were 8.4% and 6.9%, respectively. As shown in Figure 3d, sample 2 showed a high content of reducing sugars on day 0, as did sample 4, possibly caused by the addition of Daqu powder, and the reducing sugars content dropped to the lowest value on the 7th day of fermentation, then were consumed while being produced (known as simultaneous saccharification and fermentation) [28,29]. For samples 1 and 3, the content of reducing sugars showed quite a low value at the beginning of fermentation, then gradually increased during the fermentation process. This was due to the abundance of microorganisms with liquefaction and saccharification ability gradually increasing, and converting starch in the raw materials into reducing sugars. Compared with the control group, the tendency and fluctuating ranges of the physicochemical factors changed in the other three experimental groups were similar, indicating that the Baijiu fermentation process could be achieved through simulated fermentation.

### 3.2. Analysis of Flavor Compounds

The results of the physicochemical factors showed that the fermentation process of Baijiu could be simulated under laboratory conditions, while the analysis of flavor compounds was significant to judge the quality of the fermentation products, as flavor compounds play an important role in the quality of Baijiu. The flavor compounds of fermented grains were determined by GC-MS. A total of 78 flavor compounds were identified, including forty-seven esters, eight alcohols, six phenols, five alkanes, four acids, two pyrazines, two aldehydes, two ketones, and one furan in all samples during the whole fermentation process (Appendix A). There were 43 flavor compounds detected in sample 1, while samples 2, 3, and 4 contained 64, 56, and 62 flavor compounds, respectively. In addition, the relative contents of most major flavor compounds in samples 2, 3, and 4 were higher than in sample 1. For example, the relative contents of hexanoic acid, ethyl ester in samples 2, 3, and 4 were 311.18 ± 299.40 μg/kg, 545.34 ± 269.44 μg/kg, 290.05 ± 37.99 μg/kg, respectively, all being higher than that in sample 1 (124.51 ± 36.76 μg/kg); butanoic acid, ethyl ester in samples 1, 2, 3, and 4 were 131.70 ± 51.34 μg/kg, 1942.47 ± 490.64 μg/kg, 14109.75 ± 1964.85 μg/kg, 4633.89 ± 788.53 μg/kg, respectively; octanoic acid, ethyl ester in samples 1, 2, 3, and 4 were 92.82 ± 16.18 μg/kg, 110.63 ± 14.00 μg/kg, 304.11 ± 38.05 μg/kg, 289.13 ± 25.31 μg/kg, respectively (Figure 4 and Appendix A). This indicated that the microbial combination could enhance the stable production of flavor compounds. The heatmap analysis of the changes of flavor compounds during the fermentation process showed that many flavor compounds in the samples increased significantly with prolongation of the fermentation time (Appendix A). However, the changed tendencies of some flavor compounds were different, such as 2-heptanone, hexadecane, methyl caproate, and benzaldehyde in sample 1 were only detected on day 0 with very low contents, and were almost not identified with prolonged fermentation (Appendix A). These flavor compounds were possibly brought into the fermented grains by the raw materials, and no microorganisms could produce these flavor compounds during the fermentation process of the control group. 2,3,5,6-Tetramethylpyrazine is generally considered as a functional factor in Baijiu, with nutty, earth, cocoa, peanut, coffee, and asparagus aromas [30], and it was only identified in sample 2 (Appendix A) and sample 4 (Appendix A). Therefore, it is possible that 2,3,5,6-tetramethylpyrazine was introduced to the fermentation system by Daqu. The content of 2,3,5,6-tetramethylpyrazine gradually increased in the early and middle stages of fermentation, but decreased in the middle and late stages of fermentation. This indicates that the fermented samples contained microorganisms capable of the synthesis and degradation of 2,3,5,6-tetramethylpyrazine [27,30,31]. In addition, there were several middle-chain fatty acid esters, such as hexanoic acid, ethyl ester; hexanoic acid, butyl ester; octanoic acid, ethyl ester; and octanoic acid, methyl ester in sample 3 (Appendix A) and sample 4 (Appendix A), and their relative contents increased during the fermentation period (0–28 days), but decreased slightly on day 42. Studies have reported that one of the synthetic pathways of medium chain fatty acids during the fermentation of Baijiu is an esterification reaction using acid and alcohol as the substrates [2,4]. This suggests that the amount of precursors of the esters caproic acid and octanoic acid might be insufficient in the late fermentation stages, while the content of other acids was higher, and led to the reverse reaction of ester synthesis [2,32].

Cluster analysis is a classification method and can be used to automatically classify a batch of sample data according to their intimate nature without prior knowledge [33]. To investigate the similarity and divergence of the flavor compound compositions of each sample, heatmaps of the flavor compounds of the final fermented grains were clustered. The results of the heatmap analysis revealed that samples 2, 3, and 4 were closer in terms of metabolite profiles of fermented grains compared to sample 1 (Figure 4). Cluster analysis revealed that the flavor compounds of fermented grains on day 42 can be divided into four clusters, namely cluster I, cluster II, cluster III, and cluster IV (Figure 4). In cluster III, the flavor compounds in sample 4 showed the highest relative contents, followed by sample 3 and sample 2, including 3-methyl-1-butanol, phenylethyl alcohol, butanedioic acid, diethyl ester; octanoic acid, ethyl ester; decanoic acid, ethyl ester; benzoic acid, ethyl ester; 3-phenylpropionic acid ethyl ester; tetradecanoic acid, 13-methyl-, ethyl ester; (Z)-ethyl pentadec-9-enoate, dodecanoic acid, ethyl ester; tetradecanoic acid, ethyl ester; 9-hexadecenoic acid, ethyl ester; heptadecanoic acid, 15-methyl-, ethyl ester; pentadecanoic acid, ethyl ester; 9,12-octadecadienoate, propyl ester; hexadecanoic acid, ethyl ester; 9,12-octadecadienoate, ethyl ester; and 9-octadecenoic acid, ethyl ester (Figure 4). 9,12-Octadecadienoic acid ethyl ester has the function of reducing cholesterol and blood lipids, and preventing atherosclerosis [34]. In addition, butyrate esters contribute to the flavor of Baijiu [34], for example, phenylethyl butyrate has the aroma of floral, green with a tropical winey nuance; ethyl butyrate has the aroma of sweet, apple like, fresh and lifting, ethereal; isoamyl butyrate has the aroma of wax, green apple, fruit, sweet, berry; propyl butyrate has the aroma of sweet, bubble gum and pineapple like with a light green nuance; and isobutyl butyrate has sweet, fruity, pineapple, apple, bubble gum, and tropical fruit flavors [2,35]. As for cluster I, these flavor compounds showed the highest content in sample 3, followed by sample 4, and hardly any in sample 2. These results showed that the combination of key functional microorganisms (sample 3) could realize the synthesis of important flavor compounds in Baijiu, and the use of the functional microbial combination in place of some Daqu (sample 4) could not only give full play to the function of Daqu, but also promoted the stable production of many more flavor compounds together with the functional microbial combination.

### 3.3. Succession of Microbial Communities and Microbial Interactions

Flavor compounds play an important role in the quality of Baijiu, and microorganisms are the main contributors to the synthesis of flavor compounds, and show dynamic changes during the process of fermentation [29]. Therefore, exploring the composition and dynamic succession of microorganisms, and the interaction between microorganisms, could offer basic data on how these microorganisms fluctuate and contribute to flavor compound synthesis [36,37].

A total of 192 bacterial genera were detected in all samples at the genus level (Figure 5b). A Venn analysis could reflect the similarity and specificity of microbial genera in different samples [15]. According to the Venn analysis of bacterial microorganisms (Figure 5a), there were 78 bacterial genera in sample 1, 98 bacterial genera in sample 2, 137 bacterial genera in sample 3, and 114 bacterial genera in sample 4, of which 50 bacterial genera were commonly identified in all samples. In addition, it was noteworthy that there were 12, 14, 38, and 14 specific bacterial genera in samples 1, 2, 3, and 4, respectively. There were 20 genera with a relative abundance above 1% in all samples, *Lactobacillus*, *Fictibacillus*, *Sporolactobacillus*, *Clostridium sensu stricto* 11, *Clostridium sensu stricto* 12, *Burkholderia*, *Acinetobacter*, *Weissella*, *Bacillus*, *Pediococcus*, *Kosakonia*, *Clostridium sensu stricto* 1, *Achromobacter*, *Lactococcus*, *Escherichia*, *Ochrobactrum*, *Enterobacter*, *Klebsiella*, *Leuconostoc*, and *Caproiciproducens* (Figure 5b). At the initial fermentation stage, the microbial genera that were unfavorable for fermentation (e.g., *Fictibacillus*, *Bacillus*, *Pediococcus*) showed some abundance, but then rapidly decreased during fermentation [12,38,39]. On the contrary, the abundance of *Lactobacillus* in the experimental group was low at the initial fermentation stage, but showed a high abundance in the fermentation process after 7 days. Compared with all samples, *Lactobacillus* was predominant both in sample 2 on the 28th day (84.9%) and in sample 4 on the 42nd day (85.5%), while its abundance reached a maximum (25.7%) on the 14th day in sample 3 and declined gradually in the middle and late stages of fermentation. Studies indicate that *Lactobacillus* grows rapidly in a high acidity environment, in accordance with the phenomenon that the acidities of sample 2 and sample 4 increased more rapidly than that of sample 3 [40,41]. Therefore, *Lactobacillus* quickly occupied a dominant position and became the main bacterial genus at the end of fermentation in sample 2 and sample 4. This confirms that the acidity of fermented grains plays an important role in the succession of microbial communities. At the same time, the abundance of *Sporolactobacillus* in sample 1 and sample 3 gradually increased and accounted for the highest proportion of bacteria at the end of fermentation (45.8% and 60.16%, respectively). Studies have shown that *Sporolactobacillus* is one of the most dominant microorganisms in high-quality pit mud, and plays an important role in the production of flavor compounds in Baijiu [42].

In addition, the bacterial genera *Clostridium*, *Burkholderia*, and *Caproiciproducens*, in the designed microbial combination, displayed a high abundance in the fermentation system, which indicated that the added functional microorganisms could colonize the fermentation system and play a role in the fermentation process. For *Clostridium*, *Clostridium sensu stricto* 12, *Clostridium sensu stricto* 11, and *Clostridium sensu stricto* 1 were the dominant microorganisms, among which *Clostridium sensu stricto* 12 increased gradually in the early fermentation stage, while those of *Clostridium sensu stricto* 11 and *Clostridium sensu stricto* 1 increased and stabilized in the middle and late fermentation stages. In addition, the abundance of *Clostridium* in sample 3 was higher than that in sample 4, and sample 2 showed the lowest relative abundance, which also reflected that the amount of microorganisms initially added had a significant impact on the microbial succession in the fermentation process. The abundance of *Caproiciproducens* gradually increased in samples 3 and 4, and the highest abundance was 0.26%, in sample 3, on the 28th day.

At the genus level, 162 fungal genera were detected in all samples (Figure 5d). According to the Venn analysis of fungal microorganisms (Figure 5c), there were 108 fungal genera in sample 1, 113 fungal genera in sample 2, 84 fungal genera in sample 3, and 99 fungal genera in sample 4, of which 59 fungal genera were common to all samples. In addition, there were eleven, thirteen, four, and fourteen specific fungal genera in samples 1, 2, 3, and 4, respectively. There were nineteen genera with a relative abundance above 1% in all samples, including *Saccharomyces*, *Aspergillus*, *Trichosporon*, *Saccharomycopsis*, *Plectosphaerella*, *Thermoascus*, *Byssochlamys*, *Wickerhamomyces*, *Cladosporium*, *Thermomyces*, *Candida*, *Wallemia*, *Monascus*, *Rasamsonia*, *Issatchenkia*, *Papiliotrema*, *Kodamaea*, *Mortierella*, and *Meyerozyma* (Figure 5d). *Saccharomyces* and *Aspergillus* were the most important fungal genera in the fermentation process, and they were also the fungal microbes of the designed microbial combination. In all the samples, *Saccharomyces* grew rapidly and reached the highest abundances of 11.7%, 27.6%, 68.3%, and 89.7%, in samples 1,2,3, and 4, respectively, on the 14th day of fermentation, while the abundance decreased and remained stable in the middle and late fermentation stages. The genus *Saccharomyces* is usually recognized as one of the most important genera for ethanol production [43,44], while the production of ethanol was mostly performed in the early and middle stages of fermentation, thereafter, the abundance of *Saccharomyces* decreased in the middle and late stages. During the whole fermentation process, *Saccharomyces* showed a relatively high abundance in sample 4, followed by sample 3, sample 2, and sample 1 (Figure 5d). As for *Aspergillus*, the abundance was low in the early stage of fermentation in sample 3 and sample 4, and gradually increased and stabilized in the middle and late stages, with relative abundances of 41.9% and 26.6%, respectively (Figure 5d). In sample 2, the abundance of *Aspergillus* was slightly higher in the middle stage of fermentation than that in the late stage, but the maximum abundance was only 9.29% (Figure 5d). *Aspergillus* shows a good ability to saccharify starch, and the products could serve as the carbon sources for the growth of other microorganisms in the samples. The abundance of *Aspergillus* in sample 3 was continuously higher than that in other samples in the middle and late stages of fermentation, generating a higher content of reducing sugar (Figure 3d). *Monascus* was also added to the fermentation system, it showed a relatively high abundance in sample 3 (4.81%) and sample 4 (4.03%) on day 0, decreased to 0.58 % and 0.04% on the 14th day, and then remained stable (Figure 5d). This indicates that the *Monascus* in fermentation system was unable to occupy a relatively high abundance during the fermentation process.

During Baijiu fermentation, the interaction between microorganisms serves as one of the main forces for microbial dynamic fluctuations, and affects the stability of the microbial ecosystem [45]. The interactions between microorganisms diverged in different fermentation systems, and caused microbes to perform different metabolic pathways, resulting in the divergence of the production and stabilization of flavor compounds. However, complex interactions between microorganisms during fermentation has not been scientifically investigated. The above results show that sample 3 realized the combination of functional microorganisms for Baijiu fermentation, and that sample 4 had obvious advantages in the production and stability of flavor compounds. Therefore, in order to clearly analyze microbial interactions, we established co-occurrence networks among all the dominant microorganisms with significant interactions for three fermentation systems of the experimental group (Figure 6). We explained the characteristics of microbial interactions in different fermentation systems, so as to provide a basis for analyzing the complex interactions in the fermentation of Baijiu.

According to co-occurrence networks of microbial communities in each sample group (Figure 6a–c), we concluded that the positive interactions between microbial genera dominate over negative interactions, and the positive interactions mainly occurred between genera that were beneficial to fermentation. In sample 2, for the six genera (two bacterial genera *Lactobacillus* and *Clostridium sensu stricto* 1, and four fungal genera *Wickerhamomyces*, *Rasamsonia*, *Thermoascus*, and *Saccharomyces*), there were significant positive interactions between *Lactobacillus* and the other five microbial genera, and the remaining genera had significant positive interactions with the two adjacent microbial genera (Figure 6a). In addition, *Ochrobactrum* showed significant positive interactions with *Acinetobacter* and *Achromobacter*, and significant positive interactions were formed between *Clostridium sensu strictto* 11 and *Sporolactobacillus*. In sample 3, there was a large interaction network formed by seven bacterial genera (*Pediococcus*, *Burkholderia*, *Weissella*, *Lactococcus*, *Bacillus*, *Lactobacillus*, and *Clostridium sensu strictto* 12) and two fungal genera (*Aspergillus* and *Saccharomyces*). Among which, there were five positive interactions in the three genera of *Lactobacillus*, *Weissella*, and *Lactococcus*, while the fungal genus *Saccharomyces* showed significant positive interactions with four bacterial genera, including *Lactobacillus*, *Weissella*, *Lactococcus*, and *Clostridium sensu strictto* 12 (Figure 6b). Meanwhile, *Clostridium sensu stricto* 1, *Enterobacter*, and *Sporolactobacillus* formed a stable positive interaction among the three bacterial genera, and *Klebsiella* also showed a positive interaction with *Clostridium sensu stricto* 1. In sample 4 (Figure 6c), a stable positive interaction was formed among *Clostridium sensu stricto* 1, *Clostridium sensu stricto* 12, and *Klebsiella*, and the relationships between *Lactococcus*, *Weissella* and *Burkholderia* were similar. *Clostridium sensu stricto* 11 and *Caproiciproducens*, *Lactobacillus* and *Sporolactobacillus* also showed positive interactions. As for fungal interactions in sample 4, *Saccharomycopsis* showed significant positive interactions with *Thermoascus* and *Wickerhamomyces*, and *Wallemia* also showed a positive interaction with *Thermomyces*.

In addition, there were a few negative interactions in all of the fermentation systems. In sample 2, *unidentified* (Fungi) showed negative interactions with four microbial genera (*Enterobacter*, *Kosakonia*, *Klebsiella* and *Issatchenkia*) (Figure 6a). In sample 4, there was a negative correlation between *unidentified* (Fungi) and *Saccharomyces*, and the genus *Rasamsonia* and three bacterial genera (*Lactococcus*, *Weissella* and *Saccharomyces*) (Figure 6c). In sample 3, *Bacillus*, which has an adverse effect on fermentation, formed significant negative interactions with other genera (*Pediococcus*, *Weissella*, *Lactococcus*, *Lactobacillus*, *Clostridium sensu strictto* 12, *Aspergillus*, and *Saccharomyces*) (Figure 6b). It is worth noting that *Aspergillus* formed negative correlations with *Weissella*, *Bacillus*, *Lactobacillus* and *Clostridium sensu stricto* 12, which might be attributed to the extremely high abundance of *Aspergillus* on day 0 of fermentation in sample 3, and the abundance decreasing in the middle and late stages of fermentation. Meanwhile, other genera showed low abundances in the early stage of fermentation, and gradually increased as the fermentation progressed, and *Aspergillus* was negatively correlated with microorganisms of other genera.

The results of the co-occurrence networks between microbes of sample 3 included the core microorganisms *Clostridium*, *Caproiciproducens*, *Burkholderia*, *Aspergillus*, and *Saccharomyces* in the functional microbial combination, and also with microorganisms not added to the fermentation system, such as *Bacillus*, *Weissella*, *Lactobacillus*, *Saccharomycopsis*, *Thermoascus*, and *Thermomyces*, which might have come from the environment, raw materials, containers, and operators. The synergistic symbiosis of the dominant microorganisms promoted the process of Baijiu fermentation and the production of flavor compounds through the combination of the core functional microorganisms without Daqu in the fermentation system. Additionally, by replacing part of the Daqu with microorganisms, some interactions between sample 2 and sample 3 could be retained, and this also generated new interactions. The negative interactions of the microbial genera with high abundance such as *Aspergillus*, *unidentified* (Fungi), and *Bacillus* disappeared, which caused the changed metabolic function of the microbial communities. This might serve as the main reason for the more prominent flavor compounds in fermentation system 4.

### 3.4. Correlation Analysis of Flavor Compounds and Microorganisms

The dynamic succession and interactions of microorganisms affected the microbial metabolisms that produce the flavor chemicals of Baijiu. Therefore, exploring the relationships between the microorganisms and the formation of flavor compounds was of great significance for guiding the fermentation production of Baijiu [36]. The results indicated that bacterial microorganisms displayed a greater positive correlation with the flavor compounds of Baijiu (Figure 7a–c). In sample 2, *Clostridium sensu strictto* 12 and *Sporolactobacillus* showed very significant positive correlations with most of the flavor compounds (Figure 7a). In sample 3, *Sporolactobacillus*, *Clostridium sensu strictto* 1, *Enterobacter*, *Klebsiella*, *Clostridium sensu strictto* 11, and *Caproiciproducens* were positively correlated with almost all of the flavor compounds (Figure 7b). In sample 4, *Lactobacillus*, *Sporolactobacillus*, *Clostridium sensu stricto* 12, *Klebsiella*, *Kosakonia*, and *Clostridium sensu stricto* 1 showed significantly positive correlations with most of the flavor compounds (Figure 7c). A comprehensive analysis found that in all samples, *Sporolactobacillus* and *Clostridium* showed significant positive effects on the production of most of the flavor compounds in Baijiu, which was consistent with previously reported studies [42,46,47]. Additionally, more positive correlations were identified in sample 3 and sample 4 than in sample 2, and the added functional microorganisms *Clostridium*, *Caproiciproducens* and *Burkholderia* had positive effects on the synthesis and stability of flavor compounds (Figure 7a–c). The three dominant genera of *Clostridium* were *Clostridium sensu stricto* 1, *Clostridium sensu strictto* 11, and *Clostridium sensu stricto* 12 (Figure 5a). In sample 2, *Clostridium sensu stricto* 12 was most positively correlated with flavor compounds, followed by *Clostridium sensu stricto* 1 and *Clostridium sensu stricto* 11 (Figure 7a). In sample 3, *Clostridium sensu stricto* 1 and *Clostridium sensu stricto* 11 were most closely correlated with flavor compounds, followed by *Clostridium sensu stricto* 12 (Figure 7b). Sample 4 combined the advantages of sample 2 and sample 3, among which *Clostridium sensu stricto* 1 and *Clostridium sensu stricto* 12 showed the most obvious impact on flavor compounds, followed by *Clostridium sensu stricto* 11. The genus *Burkholderia* was positively correlated with flavor compound b5 (1-hexanol) in sample 4, and showed a significantly positive correlation with flavor compounds b1 (phenylethyl alcohol), a18 (decanoic acid, ethyl ester), a20 (hexanoic acid, ethyl ester), a19 (hexanoic acid, butyl ester), and a26 (octanoic acid, ethyl ester). While in sample 2 and sample 3, *Burkholderia* showed a positive impact on these flavor compounds, but not significant, which indicated that the microorganisms in sample 4 had a synergistic effect with *Burkholderia* and promoted the production of flavor compounds. *Caproiciproducens* had positive effects on most of the flavor compounds in all samples [42,47,48], especially in sample 3, it showed an extremely positive effect for a28 (acetic acid ethyl ester), a18 (decanoic acid, ethyl ester) (*p* < 0.001), and was positively correlated with the other 15 flavor compounds (twelve esters, one ketone, one alkane, one furan) (Figure 7b). In sample 2, the positive correlation between *Caproiciproducens* and flavor compounds was not significant, but it showed significantly positive effects on a8 (9.cis.,11. *trans*.-octadecadienoic acid, ethyl ester), f2 (6,10,14-trimethyl-2-pentadecanone), a11 (benzoic acid, ethyl ester), and c1 (2,4-di-tert-butylphenol) in sample 4 (Figure 7c). In addition, it is worth noting that in sample 4, flavor compounds b1 (phenylethyl alcohol), b3 (3-methyl-1-butanol), a18 (decanoic acid, ethyl ester), a20 (hexanoic acid, ethyl ester), a19 (hexanoic acid, butyl ester), a26 (octanoic acid, ethyl ester), h1 (benzaldehyde), a10 (gamma-nonanolactone), and f1 (2-heptanone) were positively correlated with most of the major bacterial genera such as *Clostridium sensu stricto* 12, *Klebsiella*, *Kosakonia*, *Clostridium sensu stricto* 1, *Enterobacter*, *Weissella*, *Lactococcus*, and *Burkholderia* (Figure 7c).

The negative correlations were mainly concentrated on *Acinetobacter*, *Achromobacter*, *Ochrobactrum*, *Fictibacillus*, and *Bacillus* with flavor compounds a14 (butanoic acid, ethyl ester), a12 (butanedioic acid, diethyl ester), a21 (lactic acid, ethyl ester), c2 (2-methoxy-4-methylphenol), a10 (gamma-nonanolactone), a1 ((Z)-ethyl heptadec-9-enoate), a22 (heptadecanoic acid, ethyl ester), a13 (butanoic acid, butyl ester), a2 ((Z)-ethyl pentadec-9-enoate), and e1 (butanoic acid) in sample 2, while in sample 3, only *Acinetobacter* displayed significant negative effects on several flavor compounds. In sample 4, *Acinetobacter*, *Fictibacillus*, and *Bacillus* showed negative effects on flavor compounds, but the negative correlations were not as obvious as those in sample 2. This confirmed that using the combination of functional microorganisms to replace part of the Daqu, could weaken the adverse effects of the Daqu microorganisms on the production of flavor compounds in the fermentation process, and promote the stable production of flavor compounds (Figure 7c).

A correlation analysis between the fungal microorganisms and flavor compounds showed that the contribution of fungi to flavor compounds was significantly less than that of bacteria (Figure 8a–c). In sample 2, *Byssochlamys* and *Issatchenkia* showed significantly positive correlations with most flavor compounds, and *Saccharomyces*, *Thermoascus*, *Wickerhamomyces*, and *Rasamsonia* showed positive correlations with most flavor compounds (Figure 8a). In sample 3, *Aspergillus* showed significantly positive effects on most flavor compounds, while *Cladosporium* and *Saccharomyces* showed significant positive effects on some flavor compounds (Figure 8b). In sample 4, *Aspergillus* also showed significantly positive effects on most of the flavor compounds, and *Cladosporium*, *Saccharomyces*, and *Trichosporon* showed significantly positive effects on some flavor compounds (Figure 8c). The results indicate that microorganisms common to sample 3 and sample 4 were positively associated with flavor compounds, but were different from those in sample 2. In sample 3 and sample 4, *Aspergillus* and *Saccharomyces* were strongly positively associated with the production of flavor compounds. *Aspergillus* was the predominant fungal genus in sample 2, and displayed positive correlations with most flavor compounds but the correlation was not obvious, while in sample 3, *Aspergillus* showed significantly positive effects on most flavor compounds (seventeen esters, two alcohols, one phenol, one alkane, one furan, one ketone, and 1one acid). In sample 4, although the positive significant effects were weaker than those of sample 3, *Aspergillus* still showed significantly positive correlations with many flavor compounds (seven esters, one phenol, one alcohol, and one acid). For *Saccharomyces*, in sample 2, it was significantly positively correlated with the medium and long chain fatty acid esters a23 (tetradecanoic acid, 13-methyl-, ethyl ester), a15 (butanoic acid, 3-methylbutyl ester), a11 (benzoic acid, ethyl ester), a18 (decanoic acid, ethyl ester), and a29 (dodecanoic acid, ethyl ester). In sample 3, *Saccharomyces* had positive effects on phenolic compounds, such as c2 (2-methoxy-4-methylphenol) and c3 (2-methoxy-4-vinylphenol), medium and long-chain fatty acid esters, such as a25 (pentadecanoic acid, ethyl ester), a23 (tetradecanoic acid, 13-methyl-, ethyl ester), and a29 (dodecanoic acid, ethyl ester), and the alcohols b1 (phenylethyl alcohol) and b3 (3-methyl-1-butanol). In sample 4, *Saccharomyces* showed a similar correlation with the flavor compounds as shown above for samples 2 and 3. Meanwhile, *Saccharomyces* also showed positive correlations with higher alcohols and medium and long chains fatty acid esters, such as b5 (1-hexanol), a20 (hexanoic acid, ethyl ester), a26 (octanoic acid, ethyl ester), a27 (acetic acid 2-phenylethyl ester), a8 (9.cis., 11.trans.-octadecadienoic acid, ethyl ester), and a10 (gamma-nonanolactone). Although the abundance of *Monascus* was lower than that of the other two fungal genera (*Aspergillus* and *Saccharomyces*), it also showed a contribution to most of the flavor compounds in all samples, especially in sample 4, *Monascus* was positively correlated with a20 (hexanoic acid, ethyl ester) (Figure 8c).

The results of negative correlation analysis between fungal microorganisms and flavor compounds showed that *Trichosporon* in sample 2 had a significant inhibitory effect on most flavor compounds. Differently from sample 2, in sample 3 and sample 4, *Trichosporon* showed positive correlations with flavor compounds a19 (hexanoic acid, butyl ester), a20 (hexanoic acid, ethyl ester), a26 (octanoic acid, ethyl ester), b1 (phenylethyl alcohol), and b3 (3-methyl-1-butanol). However, in sample 4, some fungal genera showed an inhibitory correlation with flavor compounds, such as *Wallemia*, *Thermomyces*, *Papiliotrema*, and *Rasamsonia*. The abundances of these fungal genera were not high during the fermentation process.

Esters are the most important flavor compounds in Baijiu, among which hexanoic acid, ethyl ester; butanoic acid, ethyl ester; acetic acid ethyl ester; lactic acid, ethyl ester; and octanoic acid, ethyl ester are the main flavor compounds in Strong-flavor baijiu, with fruity, pineapple-flavor, sweet, and floral aromas, and significantly affect the quality of Strong-flavor Baijiu [2,45]. The results of redundancy analysis (RDA) show the influence of the common microorganisms in the four fermentation systems on the main flavor ester compounds, and also reflect the main divergences between different samples. Therefore, an analysis of functional microorganisms and major ethyl esters was performed using RDA. The bacterial-microbial RDA found that the interpretation rate of the two axes was 86.51%, which indicated that microorganisms were closely related with the production of important esters (Figure 9a). The results showed that in sample 2, *Lactobacillus*, *Cloacibacterium*, and *Delftia* displayed only weakly positive correlations with acetic acid, ethyl ester and lactic acid, ethyl ester. In sample 3, there was a strong positive correlation between butyric acid, butanoic acid, ethyl ester; hexanoic acid, ethyl ester; octanoic acid, ethyl ester and the bacteria *Sporolactobacillus*, *Clostridium sensu stricto* 1, *Clostridium sensu stricto* 13, *Sarcina*, and *Anaerosporobacter*. In sample 4, *Lactobacillus*, *Cloacibacterium*, *Delftia*, and *Anaerosporobacter* showed strongly positive correlations with butanoic acid, ethyl ester; hexanoic acid, ethyl ester; octanoic acid, ethyl ester; acetic acid ethyl ester; and lactic acid, ethyl ester. This indicated that the fermentation by the combination of functional microorganisms made good contributions to the medium-chain fatty acids and ethyl esters, while the microorganisms in Daqu showed contributions to the short chain fatty acid ethyl esters, and the use of the functional microbial combination to replace part of the Daqu could combine the advantages of the two fermentation systems, showing a strongly positive correlation with the synthesis of both short- and medium-chain fatty acid esters.

The RDA of fungal microorganisms found that the explanation rate of the two axes reached 94.9%, which indicates that fungi show close correlations with the production of flavor esters (Figure 9b). The results show that all of the samples are clustered together during the early and middle stages of fermentation, while there are significant differences in the middle and late stages of fermentation. Positive correlations are shown between *Saccharomyces*, *Cephalotrichum*, and *Chaenothecopsis* and hexanoic acid, ethyl ester; butanoic acid, butyl ester; lactic acid, ethyl ester; acetic acid, ethyl ester; and octanoic acid, ethyl ester in sample 4. The abundance of *Saccharomyces* in all samples was high, and it showed a strongly positive correlation with several ethyl esters, which might be due to the fact that *Saccharomyces* mainly produced ethanol during fermentation, and ethanol served as the precursor for synthesizing the ethyl esters. In addition, there were positive correlations between the fungi *Saccharomyces*, *Colacogloea*, *Chaenothecopsis*, and *Trichosporon* and the flavor compounds butyric acid and butanoic acid, butyl ester.

In previous studies, *Saccharomyces* and *Lactobacillus* have been reported to be two of the main contributors to the formation of flavor compounds in Baijiu [49]. *Lactobacillus* is also active in transcribing genes involving the biosynthesis of flavor compounds and their precursors by Metatranscriptomic analysis [50]. In addition, *Clostridium* had been confirmed to have the ability to produce organic acids and short-chain fatty acids [46]. The results also confirmed the conclusion, that is, *Lactobacillus*, *Saccharomyces*, and *Clostridium* had strong positive correlations with the production of the main ethyl esters of Baijiu. During the fermentation process, these microorganisms could use small molecular substances to produce the precursors of ethyl esters such as ethanol, organic acids such as acetic acid, butyric acid, and hexanoic acid (Appendix A), so that the substrates are sufficient to promote the esterification reaction in the middle and late stages of fermentation [51]. In addition, some unreported microbial genera in our study, such as *Chaenothecopsis*, *Cloacibacterium,* and *Anaerosporobacter* showed obviously positive correlations with ethyl esters.

These results indicated that the dominant microorganisms in fermentation made a significant contribution to the production of flavor compounds. The RDA indicated strong correlations between the core functional microorganisms and the main flavor ethyl esters, which were more prominent in sample 3 and sample 4, could promote the formation of the flavor compounds and ensure the stability of Baijiu. Our studies confirmed that combining the core functional microorganisms was a stable way to Baijiu fermentation and promoted ester synthesis to ensure the product’s quality.

## 4. Conclusions

In this study, the combination of functional microorganisms could realize the production of Baijiu, which showed a good performance in raw material consumption and flavor chemical synthesis. In addition, the combination of functional microorganisms together with Daqu could effectively improve the stable synthesis of flavor chemicals. The symbiotic network analysis showed that the combination of microorganisms and Daqu could reduce the negative interactions between microorganisms, and the positive effects of microorganisms on the flavor compounds were enhanced. The RDA showed that functional microorganisms (*Lactobacillus*, *Saccharomyces*, *Clostridium*, *Cloacibacterium*, *Chaenothecopsis*, *Anaerosporobacter*, and *Sporolactobacillus*) showed strong positive correlations with the main flavor ethyl esters (hexanoic acid, ethyl ester; lactic acid, ethyl ester; butanoic acid, ethyl ester; acetic acid, ethyl ester; and octanoic acid, ethyl ester). The results of this work can provide a reference for a microbial combination for producing Strong-flavor Baijiu, help to reduce the dependence of the fermentation process on Daqu, and promote the batch stability of Baijiu fermentation.

## Figures and Tables

**Figure 1 foods-12-00644-f001:**
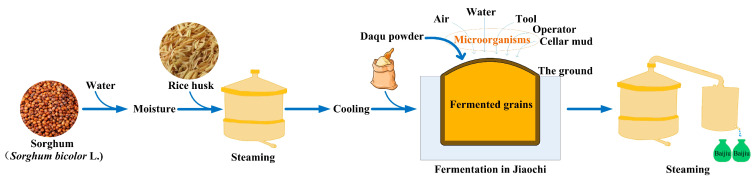
The fermentation process of Strong-flavor Baijiu.

**Figure 2 foods-12-00644-f002:**
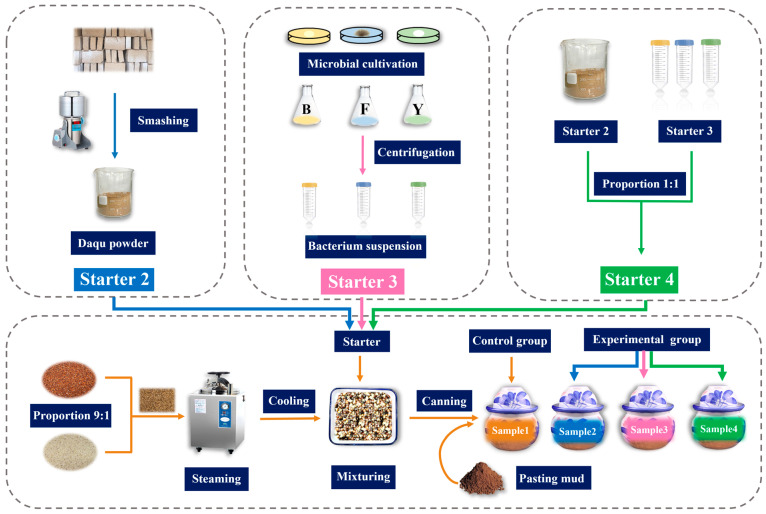
Design of simulated fermentation system. (B, bacteria; F, fungi; Y, yeast).

**Figure 3 foods-12-00644-f003:**
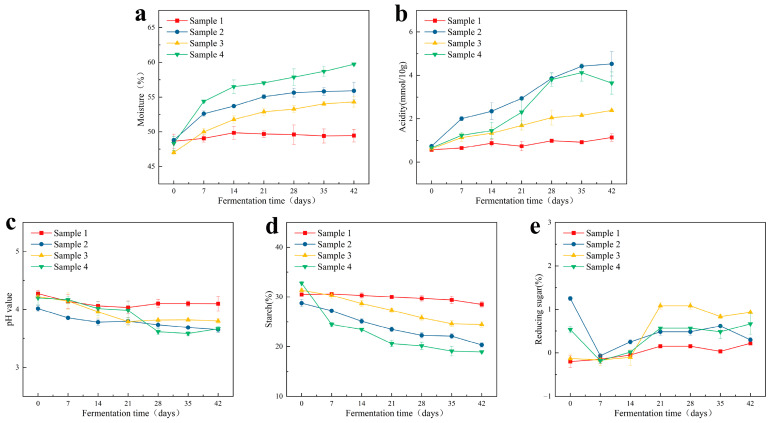
Changes of physicochemical factors in fermented grains. (**a**) Moisture, (**b**) Acidity, (**c**) pH value, (**d**) Starch content, (**e**) Reducing sugars content.

**Figure 4 foods-12-00644-f004:**
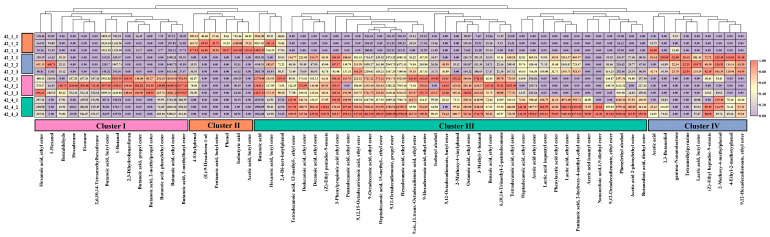
Heatmap analysis of flavor compounds in fermented grains on day 42.

**Figure 5 foods-12-00644-f005:**
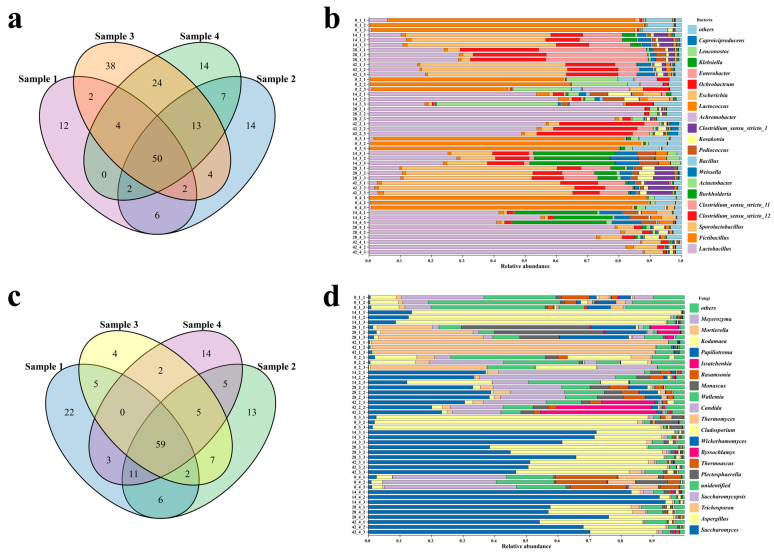
Microbial Venn analysis and community composition at the genus level. (**a**,**b**) bacteria, (**c**,**d**) fungi.

**Figure 6 foods-12-00644-f006:**

Association network of interactions between the dominant microorganisms based on correlation analysis. (**a**) Sample 2, (**b**) Sample 3, (**c**) Sample 4. The selected bacteria and fungi were calculated using Spearman’s correlation coefficient (R > 0.6, *p* < 0.05).

**Figure 7 foods-12-00644-f007:**
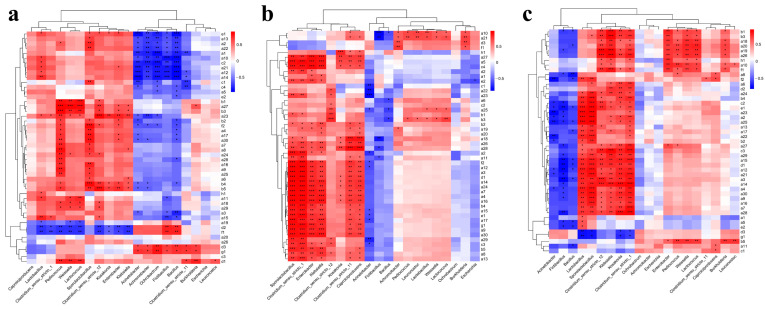
Correlation analysis of bacteria and flavor compounds. (**a**) Sample 2, (**b**) Sample 3, (**c**) Sample 4. (* *p* < 0.05, ** *p* < 0.01, *** *p* < 0.001. Each number represents a flavor compound, and the corresponding data are in Table 3.

**Figure 8 foods-12-00644-f008:**
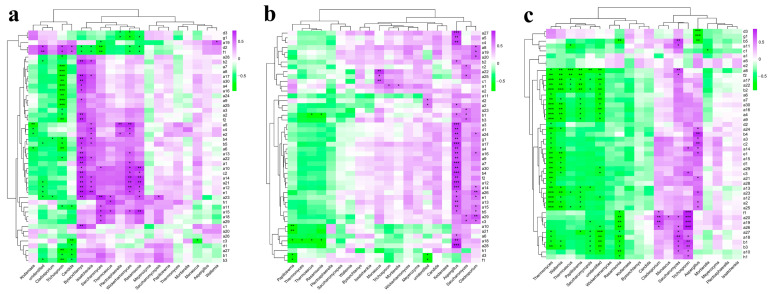
Correlation analysis of fungi and flavor compounds. (**a**) Sample 2, (**b**) Sample 3, (**c**) Sample 4. (* *p* <0.05, ** *p* <0.01, *** *p* <0.001. Each number represented a flavor compound, and the related data are listed in Table 3.

**Figure 9 foods-12-00644-f009:**
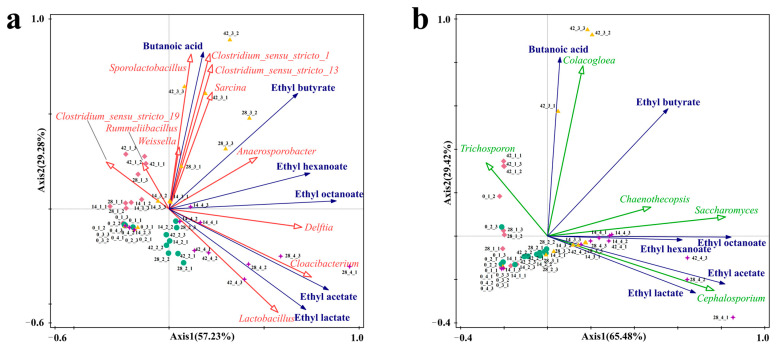
Association of microorganisms with key flavor compounds analyzed by RDA. (**a**) Bacteria, (**b**) Fungi.

**Table 1 foods-12-00644-t001:** Strains used for the microbial combination.

Microorganism	Strain	Number
Fungi	*Aspergillus niger*	CGMCC 3.4309
	*Monascus purpureus*	YJX-8
Yeast	*Saccharomyces cerevisiae*	21-9
	*Saccharomyces cerevisiae*	22-1
	*Saccharomyces cerevisiae*	22-2
Bacteria	*Burkholderia* sp.	BJQ0010
	*Ligilactobacillus acidipiscis*	JN-10-1-2
	*Clostridium beijerinckii*	BCCB2-7-1
	*Clostridium tyrobutyricum*	Gm-2-1
	*Clostridium butyricum*	JG-2-1
	*Caproiciproducens* sp.	BCJD-1
	*Clostridium kluyveri*	BJN0002

**Table 2 foods-12-00644-t002:** Details and abbreviation of the samples.

Sample Number	The Date of Samples	Sample Number for Sequencing in Triplicate
0_1	The 0 day of fermented grains of sample 1	1_0_1, 1_0_2, 1_0_3
0_2	The 0 day of fermented grains of sample 2	2_0_1, 2_0_2, 2_0_3
0_3	The 0 day of fermented grains of sample 3	3_0_1, 3_0_2, 3_0_3
0_4	The 0 day of fermented grains of sample 4	4_0_1, 4_0_2, 4_0_3
14_1	The 14th day of fermented grains of sample 1	1_14_1, 1_14_2, 1_14_3
14_2	The 14th day of fermented grains of sample 2	2_14_1, 2_14_2, 2_14_3
14_3	The 14th day of fermented grains of sample 3	3_14_1, 3_14_2, 3_14_3
14_4	The 14th day of fermented grains of sample 4	4_14_1, 4_14_2, 4_14_3
28_1	The 28th day of fermented grains of sample 1	1_28_1, 1_28_2, 1_28_3
28_2	The 28th day of fermented grains of sample 2	2_28_1, 2_28_2, 2_28_3
28_3	The 28th day of fermented grains of sample 3	3_28_1, 3_28_2, 3_28_3
28_4	The 28th day of fermented grains of sample 4	4_28_1, 4_28_2, 4_28_3
42_1	The 42nd day of fermented grains of sample 1	1_42_1, 1_42_2, 1_42_3
42_2	The 42nd day of fermented grains of sample 2	2_42_1, 2_42_2, 2_42_3
42_3	The 42nd day of fermented grains of sample 3	3_42_1, 3_42_2, 3_42_3
42_4	The 42nd day of fermented grains of sample 4	4_42_1, 4_42_2, 4_42_3

**Table 3 foods-12-00644-t003:** ID number information for flavor compounds.

Number	Flavor Compounds	Number	Flavor Compounds	Number	Flavor Compounds
a1	(Z)-Ethyl heptadec-9-enoate	a17	Heptadecanoic acid, 15-methyl-, ethyl ester	b3	3-Methyl-1-butanol
a2	(Z)-Ethyl pentadec-9-enoate	a18	Decanoic acid, ethyl ester	b4	1-Butanol
a3	3-Phenylpropionic acid ethyl ester	a19	Hexanoic acid, butyl ester	b5	1-Hexanol
a4	9,12,15-Octadecatrienoic acid, ethyl ester	a20	Hexanoic acid, ethyl ester	c1	2,4-Di-tert-butylphenol
a5	9,12-Octadecadienoate, butyl ester	a21	Lactic acid, ethyl ester	c2	2-Methoxy-4-methylphenol
a6	9,12-Octadecadienoate, ethyl ester	a22	Heptadecanoic acid, ethyl ester	c3	2-Methoxy-4-vinylphenol
a7	9,12-Octadecadienoate, propyl ester	a23	Tetradecanoic acid, 13-methyl-, ethyl ester	c4	4-Ethylphenol
a8	9.cis.,11. trans.-Octadecadienoic acid, ethyl ester	a24	Tetradecanoic acid, ethyl ester	d1	Eicosane
a9	9-Hexadecenoic acid, ethyl ester	a25	Pentadecanoic acid, ethyl ester	d2	Hexadecane
a10	gamma-Nonanolactone	a26	Octanoic acid, ethyl ester	d3	Pentylcyclopropane
a11	Benzoic acid, ethyl ester	a27	Acetic acid 2-phenylethyl ester	e1	Butanoic acid
a12	Butanedioic acid, diethyl ester	a28	Acetic acid ethyl ester	e2	Isobutyric acid
a13	Butanoic acid, butyl ester	a29	Dodecanoic acid, ethyl ester	f1	2-Heptanone
a14	Butanoic acid, ethyl ester	a30	Hexadecanoic acid, ethyl ester	f2	6,10,14-Trimethyl-2-pentadecanone
a15	Butanoic acid, 3-methylbutyl ester	b1	Phenylethyl alcohol	g1	2,3-Dihydrobenzofuran
a16	9-Octadecenoic acid, ethyl ester	b2	Isobutyl alcohol	h1	Benzaldehyde

## Data Availability

Data are contained within the article. All the data generated for this study are available on request to the corresponding author.

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
