# Peer review of "Simulated Fermentation of Strong-Flavor Baijiu through Functional Microbial Combination to Realize the Stable Synthesis of Important Flavor Chemicals"

_foods, 2023, doi:10.3390/foods12030644_

Round 1

Reviewer 1 Report

The authors describe the Baijiu fermentation using a functional microbial community. An identification of the microbial diversity during fermentation and their interactions among them and the flavor compounds was also evaluated. The content of the manuscript is well-organized, easy to follow, and provides informative results useful for scientists working in this field. However, minor points should be corrected or clarified before further consideration.

1)     Abstract; general introduction in the abstract should be minimized, and the results, such as major microbes found in the products or the main flavor compounds, should be mentioned in the abstract.

2)     In the materials and methods, item 2.2, it is better to provide the references for each analytical method, particularly acidity assay and starch and reducing sugar assay.

3)     Page 5, line 154, check the font style.

4)     In the results and discussion, item 3.1, the results of the physicochemical properties of the products should be described in the order as shown in Figure 3, i.e., acidity should be described first, following pH, moisture, reducing sugar, and starch.

5)     Apart from the correlation results between bacterial or fungal genera and flavor compounds, the metabolic pathways in synthesizing such compounds by bacterial or fungal genera should be explained to clarify such positive or negative correlation.

6)     In the references, the scientific name of microorganisms should be written in italics, i.e., ref. no. 5, 6, 7, 12, 15, 16, 17, 21, 22, 23, 27, 34, 35, 40, 42, 46. Furthermore, the word “in situ and in vitro” (ref. no. 18) and the gene name (ref. no. 35) should also be italicized.

Author Response

Response to comments from Reviewer 1 point by point

 The authors describe the Baijiu fermentation using a functional microbial community. An identification of the microbial diversity during fermentation and their interactions among them and the flavor compounds was also evaluated. The content of the manuscript is well-organized, easy to follow, and provides informative results useful for scientists working in this field. However, minor points should be corrected or clarified before further consideration.

1) Abstract; general introduction in the abstract should be minimized, and the results, such as major microbes found in the products or the main flavor compounds, should be mentioned in the abstract.

Response: Thanks for this suggestion. Abstract was revised as follows.

The solid-state fermentation of Baijiu is complicated by co-fermentation of many microorganisms, and the instability of composition and abundance of microorganisms in the fermentation process leads to fluctuations of product quality, which is one of the bottleneck problems faced by Strong-flavor Baijiu industry. In this study, we established a combination of functional microorganisms for stable fermentation of main flavor compounds of Baijiu, including medium and long-chain fatty acid ethyl esters such as hexanoic acid, ethyl ester; butanoic acid, ethyl ester; octanoic acid, ethyl ester; acetic acid, ethyl ester; 9,12-octadecadienoic acid, ethyl ester; and decanoic acid, ethyl ester in the fermented grains. Studies investigated the effects of microbial combinations on fermentation from three aspects, microbial composition, microbial interactions and microbial association with flavor compounds. Results showed that the added functional microorganisms (Lactobacillus, Clostridium, Caproiciproducens, Saccharomyces, and Aspergillus) became dominant species in the fermentation system and formed positive interactions with other microorganisms, while the negative interactions between microorganisms were significantly reduced in the fermentation system contained both Daqu and functional microorganisms. RDA analysis showed that functional microorganisms (Lactobacillus, Saccharomyces, Clostridium, Cloacibacterium, Chaenothecopsis, Anaerosporobacter, and Sporolactobacillus) showed strong positive correlations with the main flavor compounds ethyl esters (hexanoic acid, ethyl ester; lactic acid, ethyl ester; butanoic acid, ethyl ester; acetic acid, ethyl ester; and octanoic acid, ethyl ester). These results indicated that it was feasible to produce Baijiu with functional microbial combination, and could promote the stable Baijiu production.

2) In the materials and methods, item 2.2, it is better to provide the references for each analytical method, particularly acidity assay and starch and reducing sugar assay.

Response: Thanks for this suggestion. We added details and cited some references as follows.

Acidity of the samples was analyzed by acid-base titration with NaOH (0.1 M) and using the phenolphthalein as indicator (endpoint of pH 8.2) [24]. The pH value of samples was investigated by pH meter [24]. Fehling reagent method was used to determine the content of starch and reducing sugar using methylene blue as indicator [25]. Operations were carried out according to relevant standard T/CBJ 004-2018.

3) Page 5, line 154, check the font style.

Response: Thanks for this suggestion. We had modified the font within this sentence.

4) In the results and discussion, item 3.1, the results of the physicochemical properties of the products should be described in the order as shown in Figure 3, i.e., acidity should be described first, following pH, moisture, reducing sugar, and starch.

Response: Thanks for this suggestion. In the results and discussion of item 3.1, we discussed in a logical order, therefore, Figure 3 was modified in order to meet the requirement that the pictures correspond to the order of the discussion. The modified Figure 3 was as follows.

Figure 3. Changes of physicochemical factors in fermented grains. (a) Moisture, (b) Acidity, (c) pH value, (d) Starch content, (e) Reducing sugar content.

5) Apart from the correlation results between bacterial or fungal genera and flavor compounds, the metabolic pathways in synthesizing such compounds by bacterial or fungal genera should be explained to clarify such positive or negative correlation.

Response: Thanks for this suggestion. We summarized the synthetic pathways of the precursors of major ethyl esters and added explanations in the revised manuscript as follows.

In previous studies, Saccharomyces and Lactobacillus were considered as two of the main contributors to the formation of flavor compounds of Baijiu [49]. Lactobacillus was also active in transcribing genes involving the biosynthesis of flavor compounds and their precursors by Metatranscriptomic analysis [50]. In addition, Clostridium had been confirmed with the ability to produce organic acids and short-chain fatty acids [46]. The results also confirmed the conclusion, that is, Lactobacillus, Saccharomyces, and Clostridium had strong positive correlations with the production of the main ethyl esters of Baijiu. During the fermentation process, these microorganisms could use small molecular substances to produce the precursors of ethyl esters such as ethanol, organic acids such as acetic acid, butyric acid, and hexanoic acid (Fig. S5), so that the substrates are sufficient to promote the esterification reaction in the middle and late stages of fermentation [51]. Our findings were consistent with above studies. However In addition, some unreported microbial genera in our study, such as Chaenothecopsis, Cloacibacterium and Anaerosporobacter showed obviously positive correlations with ethyl esters.

Figure S5.  Synthetic pathways of fatty acids acetic acid, butyric acid and hexanoic acid.

6) In the references, the scientific name of microorganisms should be written in italics, i.e., ref. no. 5, 6, 7, 12, 15, 16, 17, 21, 22, 23, 27, 34, 35, 40, 42, 46. Furthermore, the word “in situ and in vitro” (ref. no. 18) and the gene name (ref. no. 35) should also be italicized.

Response: Thanks for this suggestion. The references were modified and marked clearly in the revised manuscript.

Thank you very much for the suggestions and comments. We hope that the answers above will enable our manuscript to win your satisfaction.

We are looking forward to hearing further information from you and would like to do as required if necessary.

Reviewer 2 Report

I have reviewed the manuscript entitled “Simulated fermentation of Strong-flavor Baijiu through functional microbial combination to realize the stable synthesis of important flavor chemicals”. This manuscript lacks methodology from both the experimental design and the analytical point of view. The novelty and significance of this work should be made more explicit, especially in the abstract and conclusion sections. I recommended major revision.

Detailed remarks about the text are as follows:

Introduction: In general, the introduction needs to be adapted to the set hypotheses and objectives of the research. It is necessary to take into account that the basic hypothesis is that the composition and proportions of certain groups of flavor compounds vary depending on the fermentation system.

Accordingly, in the introduction, it is necessary to exclude all information irrelevant for this research.

 Abstract: Abstract part should be revised. General information is given in this section. The data obtained in the study should be mentioned. Brief information on increases or decreases should be provided.

Materials and Methods: Page 5 line 152: Why were samples taken at 7-day intervals? It should be explained. How was the progress of fermentation monitored? How was the end point of fermentation determined? It should be explained.

The environmental conditions (oxygen, temperature, pH, etc.) required for yeast, fungi and bacteria differ from each other. How were environmental conditions determined?

No reference source was specified for the analyzes specified in the method section.

Page 5 line 163: Extraction methods are the most important part of this study. How conditions were determined? Was it determined in the researchers' previous work? If so, please indicate references; otherwise please discuss how these conditions were chosen, as this is very important for the results.

Conclusion: The conclusion is written more like a discussion. In conclusion, it should be concisely stated how the obtained results confirmed or disputed the set hypotheses or the objectives of the research. This is not clearly visible from the conclusion conceived in this way.

I believe that the authors put a lot of effort into the implementation of this research, but unfortunately, the work has numerous shortcomings and needs to modify.

Author Response

Response to comments from Reviewer 2 point by point

I have reviewed the manuscript entitled “Simulated fermentation of Strong-flavor Baijiu through functional microbial combination to realize the stable synthesis of important flavor chemicals”. This manuscript lacks methodology from both the experimental design and the analytical point of view. The novelty and significance of this work should be made more explicit, especially in the abstract and conclusion sections. I recommended major revision.

Detailed remarks about the text are as follows:

Introduction: In general, the introduction needs to be adapted to the set hypotheses and objectives of the research. It is necessary to take into account that the basic hypothesis is that the composition and proportions of certain groups of flavor compounds vary depending on the fermentation system.

Accordingly, in the introduction, it is necessary to exclude all information irrelevant for this research.

Response: Thanks for this suggestion. We simplified and modified introduction as follows.

Baijiu is one of the national beverages in China with a long history of thousands of years and is renowned overseas [1,2]. Among them, Strong-flavor Baijiu is the domi-nant products of Baijiu industry, with sales accounting for about 70% of the market share. The raw materials for manufacturing Strong-flavor Baijiu mainly include sor-ghum, and some enterprises also use other grains as the raw materials such as rice, glutinous rice, corn or wheat [1,2]. The raw materials are moistened with water, and mixed with rice husks and steamed thereafter (Fig. 1). Then the fermented grains are cooled to room temperature, and a certain amount of Daqu powder is added and mixed evenly, and the mixtures are transferred to the container (usually called as “Jiaochi” in Chinese) for solid-state fermentation with a time period of about 30 – 6040 days [3]. Finally, Baijiu products are obtained by distillation (Fig. 1). The Jiaochi is a container for the grain fermentation with a layer of pit mud about 10 centimeter thick on the inner wall [4], which contains specific kinds of functional microorganisms (Fig. 1). After filled with grains, the Jiaochi will be covered with a layer of mud on the top to keep the fermented grains away from the air, thus creating the anaerobic conditions for solid-state fermentation (Fig. 1).

Studies indicate the metabolism of microorganisms in the solid-state fermentation process play an important role in the transformation of raw materials [2]. Meanwhile, microbial metabolism can not only synthesize many kinds of flavor chemicals, but also degrade some potentially harmful substances in raw materials, so as to greatly affect the quality of products[5-7] [4-7]. Daqu, the fermentation starter of Baijiu, is rich in microorganisms, and has a significant impact on the Baijiu fermentation [8]. However, due to the traditional undeveloped manufacturing process of Daqu, the composition and abundance of microorganisms in different batches of Daqu are not completely consistent, leading to the quality divergence of Daqu [9-11], which has a great impact on the microbial fermentation of Baijiu. The production of high-quality Baijiu products shows a dependence on high-quality Daqu [12]. In addition, air, water, tools, operators and microorganisms in pit mud will participate in the fermentation process [2]. Thus Therefore, the microbial composition in the fermentation process of Strong-flavor Baijiu is relatively complex and unstable, and the instability of microbial compositions generate the quality instability of fermentation products among batches, which is one of the key problems to be solved in Strong-flavor Baijiu industry [2,13]. Therefore, whether we could control the complex microbial system and realize a relatively stable fermentation become a problem faced in this work.

Figure 1.  The fermentation process of Strong-flavor Baijiu.

The quality of Baijiu products is closely related to the composition of flavor chemicals [14]. Flavor analysis shows that the content of water and ethanol in the Baijiu product accounts for about 98%, and the left 2% are composed of flavor chemicals which determine the flavor and quality of Baijiu [1,15]. The composition of flavor chemicals in Baijiu is complex, including alcohols, aldehydes, acids, esters, ketones, nitrogen compounds, sulfur compounds, etc., of which the short chain fatty acid esters give a prominent contribution to the flavor of Baijiu [2,6]. According to the national standard (GB/T10781.1-2021), Strong-flavor Baijiu is one kind of product with esters as the main flavor chemicals, and among which fatty acid esters show relatively high concentrations and low odor threshold values. Previous studies also investigate the microbial metabolism of relevant flavor compounds. There are also studies about ester synthesis, mainly including two metabolic pathways, one is the transesterification re-action by yeast species, and the other is the esterification of acids and alcohols (mainly ethanol, produced by functional microorganisms such as Saccharomyces cerevisiae) [2,5]. The metabolic synthesis of fatty acids is closely related to bacteria, such as Clostridium spp. (e.g., C. butyricum and C. kluyveri) identified from the pit mud are closely related to the production of butyric acid and caproic acid, respectively [15,16]. Early studies found that fungi such as Monascus purpureus and Aspergillus niger isolated from Baijiu can synthesize esters through esterification with acids and ethanol as substrates [5,6,17]. According to the national standard of Strong-flavor Baijiu, the relevant functional microorganisms could be recognized as the important core microorganisms in the com-plex microbial fermentation system.

Recent studies about complex microbial co-fermentation system show that the core microbial composition can drive the whole fermentation process, and plays an im-portant role in maintaining the stability of the fermentation system, as well as affects the flavor and quality of products [13,18,19]. Driving the whole fermentation process with core microbial composition are recognized as helpful for maintaining the stability of microbial co-fermentation, and also applicable in Baijiu industry [13,20]. Our pre-vious work also proposed four criteria for identifying the core microorganisms from Baijiu, including (1) microorganisms hydrolyzing raw materials such as starch and protein, (2) microorganisms producing flavor chemicals, (3) microorganisms promot-ing the synthesis of flavor chemicals by other microorganisms, and (4) microorganisms coordinating the coexistence of other microorganisms [2]. Among these microorgan-isms, the strains hydrolyzing raw materials and producing flavor chemicals are par-ticularly important. As studies about microorganisms of Baijiu are still in progress, many functional microorganisms are just being discovered or remain unidentified be discovered or unidentified [21-23]. Therefore, based on previous works and criteria of functional microorganisms of Baijiu, a microbial combination was proposed, which included the functional fungal strains for raw material conversion and flavor ester synthesis, as well as ethanol producing yeast strains and acid producing bacterial strains, and the performance for Baijiu fermentation was investigated in this work. Results showed that the stable production of Baijiu could be basically realized by using the combination of functional microorganisms (without adding Daqu). In addition, the combination of functional microorganisms and Daqu for simulated solid-state fermentation could achieve the stable synthesis of important flavor esters in Baijiu by comparing the fermentation systems of the three experimental groups. This confirmed that the use of functional microbial combination could reduce the dependence on Daqu and improve the stability of production, which provided a new direction for reducing the cost and improving the quality of Baijiu. The microbial combination included the functional fungal strains for raw material conver-sion and flavor ester synthesis, as well as ethanol producing yeast strains and acid producing bacterial strains. In this work, three systems were designed with the control system to perform simulated fermentation. The stable synthesis of important flavor esters was achieved by a combination of Daqu with functional microbial combination, which confirmed that functional microbial combination was expected to promote the stability of Baijiu production. In addition, Baijiu production was realized through simulated fermentation of pure microbial combination alone (without adding Daqu) when compared with those of Daqu fermentation system, and reduced the dependence of Baijiu production on Daqu.

Abstract: Abstract part should be revised. General information is given in this section. The data obtained in the study should be mentioned. Brief information on increases or decreases should be provided.

Response: Thanks for this suggestion. Abstract was revised as follows.

The solid-state fermentation of Baijiu is complicated by co-fermentation of many microorganisms, and the instability of composition and abundance of microorganisms in the fermentation process leads to fluctuations of product quality, which is one of the bottleneck problems faced by Strong-flavor Baijiu industry. In this study, we established a combination of functional microorganisms for stable fermentation of main flavor compounds of Baijiu, including medium and long-chain fatty acid ethyl esters such as hexanoic acid, ethyl ester; butanoic acid, ethyl ester; octanoic acid, ethyl ester; acetic acid, ethyl ester; 9,12-octadecadienoic acid, ethyl ester; and decanoic acid, ethyl ester in the fermented grains. Studies investigated the effects of microbial combinations on fermentation from three aspects, microbial composition, microbial interactions and microbial association with flavor compounds. Results showed that the added functional microorganisms (Lactobacillus, Clostridium, Caproiciproducens, Saccharomyces, and Aspergillus) became dominant species in the fermentation system and formed positive interactions with other microorganisms, while the negative interactions between microorganisms were significantly reduced in the fermentation system contained both Daqu and functional microorganisms. RDA analysis showed that functional microorganisms (Lactobacillus, Saccharomyces, Clostridium, Cloacibacterium, Chaenothecopsis, Anaerosporobacter, and Sporolactobacillus) showed strong positive correlations with the main flavor compounds ethyl esters (hexanoic acid, ethyl ester; lactic acid, ethyl ester; butanoic acid, ethyl ester; acetic acid, ethyl ester; and octanoic acid, ethyl ester). These results indicated that it was feasible to produce Baijiu with functional microbial combination, and could promote the stable Baijiu production.

Grains fermentation is one of the important steps for producing Strong-flavor Baijiu, which di-rectly affects the quality of products. There are many kinds of microorganisms involved in grains fermentation. Microbial compositions in grains are usually unstable due to the traditional unde-veloped operations, and cause fluctuation of product quality, which is one of the bottlenecks faced by Strong-flavor Baijiu industry. The important flavor ester chemicals produced by microbial fermentation have a particularly significant impact on the quality of Baijiu products. The fluctuant concentrations of esters is one of the crucial reasons causing unstable product quality. In this work, Baijiu fermentations were performed through microbial combination and microorganisms to-gether with Daqu to investigate the effects on the synthesis of important flavor esters and fer-mentation stability. Flavor analysis showed that microbial combination together with Daqu could achieve the stable synthesis of important flavor esters, and the total amount of flavor chemicals increased steadily compared with the control group   and the fermentation group only added with Daqu. The stable synthesis of flavor esters could also be achieved only by fermentation using microbial composition, which could reduce the dependence on Daqu (with an important impact on Baijiu fermentation). Analysis of microbial abundance and microbial correlation with flavor chemicals showed that the artificial microbial system was relatively stable, and the added microbes showed closely positive correlation with the synthesis of flavor esters. This work provides the basic data reference for the stable synthesis of flavor esters in the actual production of Strong-flavor Baijiu in future, and will promote the batch stability of fermentation.

Materials and Methods: Page 5 line 152: Why were samples taken at 7-day intervals? It should be explained. How was the progress of fermentation monitored? How was the end point of fermentation determined? It should be explained.

Response: Thanks for this suggestion. The fermentation period of Strong-flavor Baijiu usually takes about 30-60 days. In the actual production process of Baijiu, the progress of fermentation is usually monitored by measuring the physicochemical factors of the fermented grains. In the preliminary experiment, we found that the physicochemical factors of the samples during 35-42 days were basically stable and the consumption of starch content reached about 10%, which indicated that the raw materials could be used during the fermentation period and finally reached a relatively steady state, Therefore, the 42nd day was determined as the end point of fermentation. After confirming the fermentation cycle, the time points we designed to take samples at 7-day intervals could represent different nodes in the early, middle and late stages of fermentation, and some previous studies also used a time interval of 7 d (Foods 2023, 12, 207. doi: 10.3390/foods12010207). Meanwhile, the changes of indicators at different time points in the results and discussion confirmed the rationality of the time interval.

The environmental conditions (oxygen, temperature, pH, etc.) required for yeast, fungi and bacteria differ from each other. How were environmental conditions determined?

No reference source was specified for the analyzes specified in the method section.

Response: Thanks for this suggestion. Baijiu fermentation was carried out in the Jiaochi, and undergone a transition from micro-aerobic to anaerobic conditions (J Agric Food Chem 2018, 66, 3179−3187. doi:10.1021/acs.jafc.8b00113). In the early stage of fermentation, fungi grow and decompose raw materials to produce small molecules for other microorganisms usage under micro-aerobic conditions. With the gradual consumption of oxygen, yeast metabolized substrate to produce ethanol, and the fermentation environment changed to strict anaerobic conditions, then acidogenic bacteria produced organic acids, which were esterified to produce flavor esters in the middle and late stages of fermentation. The initial conditions had a great influence on the fermentation. In this study, we referred to the environmental conditions in the actual fermentation to keep the initial conditions of the raw materials consistent (the pH value was kept around 4.0, the moisture was adjusted to 53%-55%, and the initial temperature was 20oC) according to previous reports (J Inst Brew 2017, 123 (1), 5-12. doi: 10.1002/jib.404), and water was continuously added to the top of the jar to maintain anaerobic conditions during the fermentation process. Additionally, we added details and cited some references as follows.

Five physicochemical factors during fermentation were detected, including moisture, acidity, pH value, starch content and reducing sugar content. The moisture of samples was determined using the gravimetric method by drying the samples at 105℃ for at least 3 h. Acidity of the samples was analyzed by acid-base titration with NaOH (0.1 M) and using the phenolphthalein as indicator (endpoint of pH 8.2) [24]. The pH value of samples was investigated by pH meter [24]. Fehling reagent method was used to determine the content of starch and reducing sugar using methylene blue as indicator [25]. Operations were carried out according to relevant standard T/CBJ 004-2018.

Page 5 line 163: Extraction methods are the most important part of this study. How conditions were determined? Was it determined in the researchers' previous work? If so, please indicate references; otherwise please discuss how these conditions were chosen, as this is very important for the results.

Response: Thanks for this suggestion. Extraction methods were determined in our previous work and references were cited as follows.

Volatile compounds in fermented grains were assayed by headspace solid-phase microextraction coupled with gas chromatography-mass spectrometry (HS-SPME-GC-MS) (TSQ 8000 Evo, Trace MS/GC, Thermo Fisher Scientific, Waltham, MA, USA) equipped with a flame ionization detector. Volatiles of fermented grains were extracted with a 50/30 μm DVB/CAR/PDMS fiber (Supelco, Bellefonte, PA, USA), and after equilibrating at 60℃ for 20 min, the fiber was used to extract volatiles for 30 min, followed by desorbing at the inlet for 5 min according to our previous studies [26,27].The initial column temperature was 50°C, and kept for 2 min, followed by increasing to 85°C at a rate of 2°C/min and kept for 5 min, and then increased to 150℃ at a rate of 5℃/min and maintained for 10 min, and finally increased to 250 ℃ at a rate of 5℃/min and kept for 20 min. The flow rate of a helium carrier gas was 1 mL/min [26]. The column was a TG-5MS column (30 m × 0.25 mm × 0.25 μm, J&W Scientific, Folsom, CA, USA). Mass spectrometry (MS) was generated with electron impact of 70 eV ionization energy and a full scan range from 30 to 400 amu [26]. Flavor chemicals were identified by matching the spectrum to the NIST05 spectrum database.

Conclusion: The conclusion is written more like a discussion. In conclusion, it should be concisely stated how the obtained results confirmed or disputed the set hypotheses or the objectives of the research. This is not clearly visible from the conclusion conceived in this way.

Response: Thanks for this suggestion. Conclusion was revised as follows.

In this study, the combination of functional microorganisms could realize the production of Baijiu, which showed good performance in raw material consumption and flavor chemical synthesis. In addition, the combination of functional microorganisms together with Daqu could effectively improve the stable synthesis of flavor chemicals. The symbiotic network analysis showed that the combination of microorganisms and Daqu could reduce the negative interactions between microorganisms, and the positive effects of microorganisms on flavor compounds were enhanced. RDA analysis showed that functional microorganisms (Lactobacillus, Saccharomyces, Clostridium, Cloacibacterium, Chaenothecopsis, Anaerosporobacter, and Sporolactobacillus) showed strong positive correlation with the main flavor ethyl esters (hexanoic acid, ethyl ester; lactic acid, ethyl ester; butanoic acid, ethyl ester; acetic acid ethyl ester; and octanoic acid, ethyl ester). Results of this work provided a reference of microbial combination for producing Strong-flavor Baijiu, helped to reduce the dependence of the fermentation process on Daqu, and promoted the batch stability of Baijiu fermentation.

The complex composition of microorganisms in Baijiu and the batch fluctuation of Daqu, coupled with the traditional undeveloped manufacturing process, led to the fermentation instability among batches of Strong-flavor Baijiu, which was a common problem in Baijiu industry. The combination of functional microorganisms together with Daqu could effectively improve the stability and of flavor chemical synthesis compared with the control group and the Daqu fermentation group, and only the combination of functional microorganisms also showed good performance in raw material consumption and flavor chemical synthesis. Results of this work provided a reference for the microbial combination for producing Strong-flavor Baijiu products, helped to reduce the dependence of the fermentation process on Daqu, and offered an important reference for promoting the batch stability of Baijiu fermentation.

I believe that the authors put a lot of effort into the implementation of this research, but unfortunately, the work has numerous shortcomings and needs to modify.

Thank you very much for the comments and suggestions. We hope that the answers above will enable our manuscript to win your satisfaction.

We are looking forward to hearing further information from you and would like to do as required if necessary.

Reviewer 3 Report

Overall, the authors did a good job describing the work and its significance. However, there are several issues that I would like to bring to the authors attention to improve the paper.

Lines 110-112: This sentence is awkward and needs to be rewritten. Is supposed to say ” … many functional microorganisms are just “being” discovered or “remain” unidentified.”?

Lines 120-123: Not completely sure what is meant here.

Line 154: The font appears to change within this sentence

Can you give some more detail into how the analysis physicochemical measurements like titration and pH were performed. What sample treatment if any was needed?

I do not think that table 2 adds anything to the manuscript. This information can more easily be described in text.

Line 214-216. I am not sure what this means, the sentence is unclear and should be rewritten.

It is not clear to me how the authors are reporting specific compound and total volatile concentrations. Were the compounds identified and quantified with external standards? The manuscript describes SPME analysis with an internal standards and identification by comparison to a database. Without determining response factors for each compound, the values reported would not be accurate. If this was not done than the relative amounts would be correct, but the authors should not be reporting these values as actual concentrations. If response factors were calculated than the methods should be updated.

Author Response

Response to comments from Reviewer 3 point by point

 Overall, the authors did a good job describing the work and its significance. However, there are several issues that I would like to bring to the authors attention to improve the paper.

Lines 110-112: This sentence is awkward and needs to be rewritten. Is supposed to say ” … many functional microorganisms are just “being” discovered or “remain” unidentified.”?

Response: Thanks for this suggestion. The sentence was rewritten as follows.

As studies about microorganisms of Baijiu are still in progress, many functional microorganisms are just being discovered or remain unidentified. be discovered or unidentified.

Lines 120-123: Not completely sure what is meant here.

Response: Thanks for this suggestion. We had modified the sentences.

Results showed that the stable production of Baijiu could be basically realized by using the combination of functional microorganisms (without adding Daqu). In addition, the combination of functional microorganisms and Daqu for simulated solid-state fermentation could achieve the stable synthesis of important flavor esters in Baijiu by comparing the fermentation systems of the three experimental groups. This confirmed that the use of functional microbial combination could reduce the dependence on Daqu and improve the stability of production, which provided a new direction for reducing the cost and improving the quality of Baijiu.

Line 154: The font appears to change within this sentence

Response: Thanks for this suggestion. We had modified the font within this sentence.

Can you give some more detail into how the analysis physicochemical measurements like titration and pH were performed. What sample treatment if any was needed?

Response: Thanks for this suggestion. We added details and cited some references as follows.

Acidity of the samples was analyzed by acid-base titration with NaOH (0.1 M) and using the phenolphthalein as indicator (endpoint of pH 8.2) [24]. The pH value of samples was investigated by pH meter [24]. Fehling reagent method was used to determine the content of starch and reducing sugar using methylene blue as indicator [25]. Operations were carried out according to relevant standard T/CBJ 004-2018.

I do not think that table 2 adds anything to the manuscript. This information can more easily be described in text.

Response: Thanks for this suggestion. The sample number information in table 2 appeared in the figure of the "Results and Discussion", we thought readers would get the effective information from the figure and table more clearly.

Line 214-216. I am not sure what this means, the sentence is unclear and should be rewritten.

Response: Thanks for this suggestion. The sentence was rewritten as follows.

In particular, the rising ratio of acidity in samples 2 and 4 were higher than that of sample 3, and the divergence was possibly caused by the difference of microbial community compositions (e.g., Lactobacillus could produce lactic acid, and its abundance in sample 3 was lower than those of samples 2 and 4, strongly associated with acidity), and generated varied metabolites. In particular, the rising ratio of acidity and the acidity at the fermentation end point in samples 2 and 4 were higher than that of sample 3, the divergence was possibly caused by the difference of microbial community compositions (e.g., the abundance of Lactobacillus in sample 3 was lower than those of samples 2 and 4), and generated varied metabolites.

It is not clear to me how the authors are reporting specific compound and total volatile concentrations. Were the compounds identified and quantified with external standards? The manuscript describes SPME analysis with an internal standards and identification by comparison to a database. Without determining response factors for each compound, the values reported would not be accurate. If this was not done than the relative amounts would be correct, but the authors should not be reporting these values as actual concentrations. If response factors were calculated than the methods should be updated.

Response: Thanks for this comment. We used NIST MS database to identify flavor compounds, and 4-octanol as the internal standard for relative quantification of flavor compounds, so the values reported could not represent actual concentrations of flavor compounds in the samples. In addition, we corrected the inappropriate formulation in the revised manuscript, changed “content” to “relative content” and “concentration” to “relative concentration”. And we also modified sentences as follows.

In addition, the total relative contents of flavor compounds in samples 2, 3 and 4 were 80.1 mg/kg, 182.4 mg/kg and 132.2 mg/kg, respectively, all were higher than that in sample 1 (51.3 mg/kg), indicating that Daqu or microbial combination could enhance the stable production of flavor compounds. In addition, the relative contents of most major flavor compounds in samples 2, 3 and 4 were higher than in sample 1. For example, the relative contents of hexanoic acid, ethyl ester in samples 2, 3 and 4 were 311.18 ± 299.40 μg/kg, 545.34 ± 269.44 μg/kg, 290.05 ± 37.99 μg/kg, respectively, all were higher than that in sample 1 (124.51 ± 36.76 μg/kg); butanoic acid, ethyl ester in samples 1, 2, 3 and 4 were 131.70 ± 51.34 μg/kg, 1942.47 ± 490.64 μg/kg, 14109.75 ± 1964.85 μg/kg, 4633.89 ± 788.53 μg/kg; octanoic acid, ethyl ester in samples 1, 2, 3 and 4 were 92.82±16.18 μg/kg, 110.63±14.00 μg/kg, 304.11±38.05 μg/kg, 289.13±25.31 μg/kg (Fig. 4 and Table S1). This indicated that microbial combination could enhance the stable production of flavor compounds.

Thank you very much for your suggestions. We hope that the answers above will enable our manuscript to win your satisfaction.

We are looking forward to hearing further information from you and would like to do as required if necessary.

Round 2

Reviewer 2 Report

The revised manuscript titled ‘Simulated fermentation of Strong-flavor Baijiu through functional microbial combination to realize the stable synthesis of important flavor chemicals’ has been reviewed. I am glad to see that the authors responded meticulously to the questions and suggestions. The general impression of the revised paper is that it has been improved by the authors with high attention to the number of important details.

Best regards